# Dutch Translation of the Yost Self-Report Lower Extremity Lymphedema Screening Questionnaire in Women

**DOI:** 10.3390/cancers16132396

**Published:** 2024-06-28

**Authors:** Tina Decorte, Charlotte Van Calster, Caren Randon, Vickie Van Besien, Mathilde Ketels, Luc Vanden Bossche, Mieke De Schryver, Chris Monten

**Affiliations:** 1Department of Physical Medicine and Rehabilitation and Clinic for Lymphatic Disorders, Ghent University Hospital, 109000 Ghent, Belgium; 2Department of Rehabilitation Sciences, University of Leuven and Lymphoedema Center, Leuven University Hospital, 493000 Leuven, Belgium; 3Department of Thoracic and Vascular Surgery and Clinic for Lymphatic Disorders, Ghent University Hospital, 109000 Ghent, Belgium; caren.randon@uzgent.be; 4Department of Physical Therapy and Motor Rehabilitation, Ghent University, 109000 Ghent, Belgium; 5Department of Physical Medicine and Rehabilitation, Ghent University Hospital, 109000 Ghent, Belgium; 6Clinic for Lymphatic Disorders, Ghent University Hospital, 109000 Ghent, Belgium; 7Department of Radiation Oncology and Clinic for Lymphatic Disorders, Ghent University Hospital, 109000 Ghent, Belgium

**Keywords:** Lymphedema, self-reported questionnaire, gynecological cancer, Dutch translation

## Abstract

**Simple Summary:**

Lymphedema is a common complication following gynecological cancer treatment, particularly affecting the lower limbs. However, self-reported data on lymphedema detection remain scarce. To address this gap, Yost developed the Lower Extremity Lymphedema Screening Questionnaire (LELSQ) specifically for women. The LELSQ is a straightforward and user-friendly tool designed for the early identification of lower extremity lymphedema (LEL). The validation of the questionnaire in Dutch was crucial for a larger trial called “The Gynolymph”, since there were no validated questionnaires in Dutch for the detection of early LEL. This trial aims to enroll four hundred patients who will utilize the screening tool as part of a patient-reported assessment. By doing so, the trial seeks to detect the early development of lymphedema after cancer treatment. To ensure unbiased results, the study team translated and adapted the lymphedema questionnaire to Dutch. The cross-sectional survey conducted during this process demonstrated high internal consistency, test–retest reliability, and validity. Additionally, face and content validity were confirmed, allowing for the implementation of the questionnaire in the early detection of LEL among Dutch-speaking women.

**Abstract:**

Background: Validated questionnaires of self-reported LEL are important in the assessment and diagnosis of LEL. The aim of this study was to validate and translate a Dutch version of the screening questionnaire, the LELSQ developed and validated by Yost et al. Methods: We tested the questionnaire on a group of healthy women and a group of patients diagnosed with LEL. The translation was carried out using the forward and backward method from English to Dutch. Statistical analyses: SPSS (IBM corp, Armonk, New York, NY, USA) version 28.0.1.0 (001) was used for statistical analysis in the process of validation. The internal consistency was assessed by determining Cronbach’s alpha. The reliability was tested by test–retest reliability. The validity was determined by ROC analysis, and content and face validity were evaluated. Results: The internal consistency score in both groups had a strong value (0.83 to 0.90). The test–retest reliability was also strong in both groups. Face and content validity showed the LELSQ is an easy, understandable questionnaire that is not too time-consuming in the early detection of LEL. The ROC analysis showed an AUC value of 0.93, indicating strong validity. Conclusions: The validated Dutch translation showed high values for internal consistency, test–retest reliability, and validity, which allows us to implement the questionnaire in the early detection of LEL after gynecological cancer treatment.

## 1. Introduction

Lower extremity lymphedema (LEL) is a serious condition characterized by swelling, pain, and functional impairment, often impacting the psychosocial well-being of these patients. LEL is caused by a reduced transport capacity of the lymphatic system leading to an accumulation of protein-rich lymph fluid in the subcutaneous tissue. The occurrence of LEL can be either sudden or gradual. In high-income countries, it mainly occurs secondary to cancer treatment, with presentation in one or both limbs [1]. Since there exists no consensus on a uniform diagnosis of LEL, there is a broad range in the rate of incidence of LEL after cancer treatment: from 0 to 70% [2,3,4]. Lymphedema is a chronic condition that can be treated to prevent progression but cannot be cured. The earlier it is detected, the higher the chance that skin changes, chronic wounds, erysipelas, and functional impairment, all impacting quality of life, can be prevented [5,6]. These stresses demonstrate to what extent early detection is imperative. Secondary LEL usually develops several months to years following the completion of cancer therapy, with peak incidence after 6–12 months [7,8]. Disease-related patient-reported outcome measures (PROMs) play an important role here. Indeed, patient-reported symptoms are often the first indication of LEL. Furthermore, they guarantee a patient-centered evaluation in the function of disease detection [9]. Yost et al. developed and validated a screening questionnaire for the detection of lymphedema at the level of the lower limbs [5]. This screening tool can be easily implemented in a post-therapy lymph diary to detect LEL at an early stage. However, as for now, there is no specific questionnaire available for screening lymphedema in the lower limbs in Dutch. We currently possess the validated Dutch-language Lymphedema Functioning, Disability, and Health Questionnaire (lymph-ICF). This questionnaire serves as a reliable measurement tool to assess the quality of life for patients with lymphedema in the lower limbs but not for the early detection of LEL [10]. The primary objective of our study was to validate and translate a Dutch version of the validated Lower Extremity Lymphedema Screening Questionnaire (LELSQ) developed by Yost. We conducted this validation process within a cohort of Dutch-speaking women, including both those without and those with LEL. Proper validation of the questionnaire is important to ensure the instrument can be used correctly. Validation comprises different analyses, which means that the translated LELSQ should score high in reliability and validity. The reliability is an indicator for the consistency of the results, which can be tested in terms of internal consistency and test–retest reliability. Internal consistency is an indicator that reflects if all items of the questionnaire measure the same construct [11,12]. The test–retest reliability indicates if the responses remain consistent within a short period of time.

## 2. Materials and Methods

All subjects gave their informed consent for inclusion before they participated in the study. Ethical approval for this cross-sectional research was obtained by the Ethical Committee of Ghent University Hospital (ONZ-2022-0224 and ONZ-2023-0529). The deidentification of all data was performed before they were used for analysis. This study complies with the Declaration of Helsinki. This study is a sub-study of a trial that has been registered in ClinicalTrial.gov NCT05469945. For this study, the LELSQ was used, as developed, and validated by Yost et al., upon written approval for copyright usage [5]. They developed a screening questionnaire, the LELSQ, with the goal of a low-threshold self-reported detection tool facilitating the faster diagnosis of LEL, with specific attention to differentiate between adiposity and lymphedema in women with a BMI ≥ 30 kg/m². Obesity is a common comorbidity in LEL, where signs of lymphedema such as swelling, heaviness, or discomfort may be masked by adiposity. In the original article of Yost, they validated the questionnaire in obese and non-obese patients based on a prospective analysis using data from 127 women undergoing cancer therapy. They were divided into two groups: those with BMI < 30 (51 women) and obese women with BMI ≥ 30 (76 women). Based on statistical analysis and a screening of the questions by an expert panel, the final screening questionnaire was obtained, consisting of two parts. The first part included 13 questions about swelling and symptoms, and the second part included figures reporting swelling at the lower limb with 5 additional questions. The sum of the 13 items (score range = 0–52) with a cut-off of more than 5 points was found positive for screening, with a sensitivity and specificity for all participants in the study of 95.5% and 86.5%, respectively, and for the subgroup of obese women, 94.8% and 76.5% [5]. Since the early diagnosis of LEL is challenging after cancer treatment, we used the 13-item questionnaire to translate and validate in Dutch (Appendix A).

### 2.1. Translation Process

Before implementing this screening questionnaire for Dutch-speaking women, it is crucial to validate the translation. A commonly used translation methodology for translating medical research questionnaires is the ‘forward–backward translation’ process. In this approach, the questionnaire is initially translated from English into the target language and subsequently back–translated to English [12,13]. The LELSQ was translated following the international guidelines as described by the World Health Organization [13,14]. Initially, the questionnaire underwent translation into Dutch by four native Dutch-speaking translators with English proficiency levels ranging from B2 to C1 (forward-translation) [15]. These translators were experts in health and lymphatic disorders. The translation process was conducted independently. Suggestions for the translation were collected, followed by a consensus-building discussion among the translators and an independent reviewer to address any variations in conceptual understanding and semantic interpretation. An independent near-native English-speaking individual (proficient at level C2) conducted the back-translation. This method enables the research team to assess the extent to which the translation aligns with the source items [15]. The back–translation revealed only a minor grammatical difference with the original (‘I have a swollen of’ instead of ‘I have swelling’), which was accepted as an insignificant difference.

The Dutch version was first tested on volunteers to evaluate reliability. In a second phase, this version was tested on a patient population with objectified lymphedema.

### 2.2. Study Population

The Dutch questionnaire was tested on a heterogenous group of healthy women (group 0), and a group of patients diagnosed with LEL (group 1).

Due to the diversity of questionnaire types, there are no universally applicable rules regarding the required sample size for questionnaire validation [16]. While larger sample sizes are generally preferable, there is no one-size-fits-all rule for determining the necessary sample size for questionnaire validation. Additionally, considering the respondent-to-item ratios can provide further justification for a larger sample size when needed. The number of volunteers/patients needed per group was calculated through sample size analysis, imposing a power of at least 90%, based on the formula by Bonett [17]. There are 13 items in the questionnaire that needs to be assessed for the reliability of its measurements [18]. According to this formula, the calculated sample size we needed was a minimum of 13 persons. Another guideline for the respondent-to-item ratio is the 5:1 rule [11]. The rule suggests that for every item in the questionnaire, we aim for at least 5 respondents. A minimum sample size of 65 respondents is required for the 13-item questionnaire based on the recommended respondent-to-item ratio of 5:1. As a test–retest was foreseen within 2 weeks after the first completion, the number of participants was increased to 100 to compensate for potential drop-out. In this study, group 0 consists of a heterogeneous group of women without LEL, while group 1 comprises patients diagnosed with LEL. After collecting signed informed consent forms, participants received both oral and written information about the study. The LELSQ was administered on two separate occasions, with a two-week interval between assessments. Following this period, all participants received an email containing a link to complete and return the questionnaire, allowing researchers to assess the test–retest reliability.

### 2.3. Statistical Analyses 

For statistical analysis, we utilized SPSS (IBM Corp, Armonk, New York, NY, USA) version 28.0.1.0 (001). The analysis was conducted separately for each group and for the total group. Additionally, all statistical tests were validated by the biostatistics unit at the University of Ghent.

#### 2.3.1. Internal Consistency

In a scale comprising several items, it is essential to assess whether these items measure the same underlying construct. This process helps determine the internal consistency of the scale. In other words, it estimates how reliable the responses from the questionnaire are [17]. This is examined by means of Cronbach’s alpha analysis. This analysis is based on the premise that each item in the scale is sufficiently correlated with each other item in the same scale. The value of Cronbach’s alpha ranges from 0 to 1, with higher values implying the items are measuring the same thing [18]. A score below 0.40 was interpreted as weak, a score between 0.40 and 0.74 was interpreted as moderate, a score between 0.75 and 0.90 was interpreted as strong, and a score above 0.90 was interpreted as very strong [17,18]. Additional testing was conducted to verify if we obtained a higher Cronbach alpha’s score when an item was removed from the questionnaire.

#### 2.3.2. Test–Retest Reliability

Test–retest reliability is the extent to which the scores of a questionnaire are stable over time. To examine test–retest reliability, the questionnaire must be filled out on two separate occasions, with an interval that is sufficiently short to assume the underlying condition did not change but long enough so that participants do not remember their previous answers. To measure test–retest reliability, the two-way mixed effects interclass correlation coefficient (ICC) was used. The ICC can theoretically vary between 0 and 1.0, where an ICC of 0 indicates no reliability and an ICC of 1.0 indicates perfect reliability [19]. The closer the coefficient is to 1.0, the higher the reliability. A coefficient above 0.7 is considered to be good, and a coefficient higher than 0.8 is considered to be excellent. 

#### 2.3.3. Criterium Validity

Validity refers to the degree a questionnaire has consistently measured what needed to be measured. Criterium validity is the degree to which the scores of the questionnaire reflect a gold standard (i.e., clinical diagnosis of LEL by a health care professional). The area under a receiver operating characteristic (ROC) curve (the AUC) is used as a measure of the performance of a screening test [20]. ROC analysis can be used as an alternative to other validity analyses [21].

#### 2.3.4. Face and Content Validity

The face and content validity of the LELSQ were measured at the time the participants signed the informed consent by an additional questionnaire with 3 questions to fill out:Was the scoring system comprehensible? Yes/No.Was each question of the Dutch LELSQ understandable? Yes/No.Were all complaints related to your lymphedema questioned in the LELSQ? Yes/No.

These additional questions were also translated by the four Dutch translators, following the forward–backward translation method. If the answer to any of these questions was no, a written explanation was requested. To define face and content validity, a scoring of very good (> 90% of patients thought the questionnaire understandable and complete), good (between 75% and 90%), moderate (between 40% and 74%), and weak (<40%) was used.

The data associated with the paper are not publicly available but are available from the corresponding author on reasonable request.

## 3. Results

### 3.1. Study Population

One hundred patients were included in the study, all females. Patients of group 0 (without LEL) were recruited between September and November 2022 and between November 2023 and April 2024 at the department of Physical Medicine and Rehabilitation of Ghent University Hospital during consultation. Patients were eligible when they did not have a history of LEL or other types of edema and were native Dutch speakers. The patients of group 1 (with diagnosis of LEL) were recruited between November and December 2022 and between November 2023 and April 2024 at the Clinic of Lymphatic Disorders of Ghent University Hospital during their consultation for treatment of lymphedema. Patients were eligible when they had a diagnosis of primary or secondary LEL and were native Dutch speakers. The diagnosis of lymphedema was determined based on the clinical stage using the International Society of Lymphology (ISL) staging system. Patients were considered eligible if they had grade I, II, or III lymphedema. All patients had a lymph scintigraphy to confirm diagnosis. Among the 50 included patients, 58% had ISL grade I LEL, 42% had grade II LEL, and no patient had grade III LEL. The overall mean age in both group 0 and 1 was 45.28 years (SD 15.02), with a range of 20–83 years, and a mean body mass index (BMI) of 27 (SD 6.98), with a range of 15.5–53.3. The demographics of all participants per group are listed in Table 1.

### 3.2. Internal Consistency 

The internal consistency of the LELSQ in the non-lymphedema group showed a value of 0.84, and for the group with LEL, a value of 0.83. The internal consistency score in both groups combined had a value of 0.91 (Table 2).

The values of 0.83 to 0.91, hence, were categorized to be strong. The group without LEL had 2 questions where the answer was 0, so only 11 items were included in the Cronbach alpha analysis, omitting the 2 irrelevant questions. The questions ‘have I swelling in my buttocks’ and ‘have I swelling in my genital area’ were all answered 0 in the non-lymphedema group. Basically, this means that those questions have no added value within the investigated target group. Specifically, within the non-patient group, only 0 was answered to these questions by all participants. Therefore, it is not possible to calculate a correlation for these questions based on the formula between X and Y: covariance XY/(SDX*SDY). As covariance, variance and therefore SD are all equal, and the consequent result, zero divided by zero, is not defined. Factor analysis revealed that deleting a question has no or little impact on the Cronbach’s alpha score in both groups. Thus, the factor structure was found to be strong to very strong. 

### 3.3. Test–Retest Reliability 

Among the total group of 100 participants, 93 completed the LELSQ twice within an interval of 2 weeks. To test the test–retest reliability of the total score of the Dutch LELSQ, its intraclass correlation coefficients (ICCs) and its 95% confidence interval were calculated. In the Dutch translation of the LELSQ, we have an ICC of 0.89 in the group without LEL and 0.83 in the LEL group, meaning we have a strong result for test–retest reliability. Bland–Altman plots (Figure 1) showed no systematic differences between the test and retest in both groups (P = NS). 

However, the variation in the LELSQ increased with higher LELSQ scores (Figure 1C). Subgroup analyses showed that reliability was drastically better in patients with initial LELSQ scores lower than 15 compared to patients with LELSQ scores higher than 15. The cut-off score of 15 was chosen since the LELSQ is reliable as long as the score remains below 15. At higher scores, there is clearly more variation. Additionally, no correlation was found between LELSQ values and the severity of the lymphedema as assessed by a physician.

### 3.4. Criterium Validity

For criterium validity as part of the validation process, an ROC analysis was performed on the LELSQ for patients with or without lymphedema. So far, no other patient questionnaire is available regarding the early detection of LEL in Dutch. The purpose of the questionnaire is to detect lymphedema even before clinical symptoms appear. The AUC determines the accuracy rate of the test in discriminating the results of patients without LEL versus those of patients with LEL. ROC analysis showed an AUC value of 0.93, indicating strong validity (*p* < 0.001) (95% CI: 0.86 to 0.97). The Youden index was 0.75, with a cut-off value of >6. This resulted in a sensitivity of 90.5% and a specificity of 84%.

### 3.5. Face and Content Validity

The results of the face and content validity analysis are presented in Table 3. 

More than 95% of the participants found the questionnaire understandable and easy to score. In the LEL group, 13 patients expressed that not everything about their LEL was adequately surveyed. Additional information to describe missing information was asked for in such cases. One person could not really indicate anything but had answered the question negatively. Another person indicated that there were no questions about lymphedema at the level of the arms and neck, while it was clearly indicated that it was a scale only covering LEL and therefore not applicable to the whole body. Another participant indicated that there was no question about the presence of bruises, while another person reported not having had a question about cold intolerance, though both symptoms are not related to LEL. Three patients mentioned that there were no questions about paresthesia in the lower limb. Two patients noticed the absence of a question about swelling of the toes or foot. One patient indicated that she felt the questionnaire did not take limb mobility into account. There was no specific question about fatigue. One patient suggested splitting the question about pain into rest pain and pain related to compression contact.

### 3.6. Influence of BMI

A Pearson correlation coefficient was used to measure the linear relationship between BMI and a higher score on the LELSQ. The calculated correlation coefficient is 0.28. This indicates a weak correlation between BMI and a higher score on the LELSQ. We also carried out an ROC analysis for the patient group with LEL and a BMI ≤ 30 and patients with LEL with a BMI ≥ 30. The AUC value in patients with a BMI ≤ 30 was 0.93 (95% CI: 0.86 to 0.98). At a threshold score of 6 on the LELSQ, the specificity was 0.89 and sensitivity was 0.72, with a Youden Index of 0.70. For patients with a BMI ≥ 30, the AUC in the ROC analysis was 0.96 (95% CI: 0.79 to 1). These values suggest a good discriminative ability.

## 4. Discussion

This study validated the translation process of the existing LELSQ questionnaire into Dutch (Appendix B). Considering that there is currently no available translation, this instrument would be valuable for evaluating Dutch-speaking patients in the early detection of secondary lymphedema after gynecological cancer treatment. To be effective as a screening tool in practice, it is not only important to be relevant as a PROM but also to be easy to use for patients and health professionals. Screening tools should be clear, user-friendly, and time-saving tools with a clear scoring system and questions that are clinically relevant to signs and symptoms. The LELSQ meets all the predefined requirements [5,6,7]. In addition to swelling, pain/discomfort, and skin texture, feelings of tension and heaviness are also questioned. This is important, as some women with LEL experience these symptoms in the absence of the typical bothersome swelling. The LELSQ is an easy, understandable questionnaire that is not too time-consuming for the early assessment of LEL in patients after gynecological cancer treatment. The initial comprehensive questionnaire by Yost et al. was developed consisting of four sets of questions. The first three different sets of questions focus on the absolute and relative extensiveness of signs or symptoms at different locations of the lower extremities and on the reporting of the degree of swelling [5]. The fourth set of items refer to the illustrated body figures, not used in our study. The questionnaire has been validated in German and Norwegian [22,23]. In a recent German translation of the LELSQ, four additional questions were added in the translated screening tool [22,23]. Three of these additional questions pertain to pitting edema. It is important to note that this sign is not specific to lymphedema alone; it also occurs in venous edema and lipedema. Consequently, we have chosen to proceed with the original 13-item questionnaire developed by Yost et al. The LELSQ is the first specific instrument to be translated to Dutch and adapted for individuals with secondary lymphedema in the early detection of lower limb lymphedema. It can be seamlessly integrated into a lymph diary to identify early lymphedema of the lower limbs in women after gynecological cancer treatment. The translated questionnaire demonstrates strong to very strong internal consistency. This validated questionnaire will play a pivotal role in a future study focused on the early detection of LEL after gynecological cancer. In the validation process, it was suggested to leave out two questions on genital edema and shorten the questionnaire for patients. We know that after gynecological laparoscopy, vulvar edema can occur as a rare complication [24]. Considering this, it is desirable to include this rare complication in the screening tool. The questionnaire can be used in research as well as in clinical settings.

The test–retest reliability of the Dutch LELSQ was strong in both groups. However, in both groups, the reliability of the LELSQ was highly dependent on the presence of lymphedema. While the LELSQ showed excellent reliability in LELSQ scores lower than 15, patients with LELSQ > 15 showed poor reliability. This confirms the statement of the authors of the LELSQ, who also reported that the screening tool cannot be used to assess the severity or evolution of LEL. Nonetheless, as the cut-off point for the diagnosis of lymphedema has been determined to be 6 in this study and 5 by the inaugural report, this lies well below the threshold for reliable values (LELSQ < 15). With a score of 6, we still have both a sensitivity and specificity of over 80%. Further research to refine and validate the cut-off score for sufficient sensitivity and specificity is needed. As such, the LELSQ remains a promising screening tool. The original report also points out that further research should be performed to subdivide groups regarding BMI. The reliability of the screening tool is lower in obese women, but these cannot currently be distinguished by the degree of obesity above a BMI of 30 kg/m². We found a weak correlation between BMI and a higher score in the total group. The diagnosis of lymphedema is challenging since signs and symptoms are often underrecognized [2]. Notably, in most women, LEL occurs in both lower limbs, and the LELSQ does not differentiate between uni- or bilateral LEL. Moreover, only the most affected limb is evaluated. It is essential to consider that participants recruited in the initial study by Yost et al. were already receiving lymphatic therapy, potentially leading to milder symptom presentation. Consequently, this could result in a lower score on the LELSQ, impacting reported sensitivity [5]. Authors should thoroughly discuss the results, interpreting them within the context of previous studies and working hypotheses. Additionally, the implications of the findings should be explored broadly, and future research directions may be highlighted.

## 5. Limitations

There were some limitations to this study. Unfortunately, one of the COSMIN criteria could not be met: due to absence of a gold standard—as of yet, there are no other Dutch validated questionnaires for the early detection of LEL—a part of the validity criteria, discriminant validity, could not be performed [25]. The Gynecological Cancer Lymphedema Questionnaire (GCLQ) is another widely used tool to detect postoperative lymphedema. However, it has not been validated in Dutch and hence could not be used as a gold standard [26]. Another limitation is the bundling of pain and discomfort into one question in the LELSQ, potentially leading to confusion. Pain is defined as an unpleasant sensory or emotional experience associated with (possible) tissue damage. Discomfort, on the other hand, may be more indicative of a milder pain or may encompass a broad spectrum of health-related complaints [27,28]. Sensitivity may improve with the separation of these items.

## 6. Conclusions

Currently, there is limited evidence regarding the early detection of lower limb lymphedema after cancer treatment. Yost at al. developed the LELSQ, a straightforward and user-friendly screening tool for identifying lymphedema in women after gynecological cancer treatment. The validated Dutch translation of the LELSQ demonstrated strong internal consistency, test–retest reliability, and validity, including face and content validity. This validation allows us to effectively implement the questionnaire for the early detection of LEL. Specifically, we can integrate the translated LELSQ into a lymph diary app, which women can complete after their cancer treatment. If patients indicate a score of 6 or higher on the LELSQ, confirmed through auto-re-evaluation, we can identify those at higher risk of developing LEL. As recommended by the LELSQ authors, it is desirable to invite these women for early clinical evaluation and radiological detection of LEL. Detecting LEL early enables us to delay or even prevent the deterioration of lymphedema, potentially positively impacting the quality of life for these women [29].

## Figures and Tables

**Figure 1 cancers-16-02396-f001:**
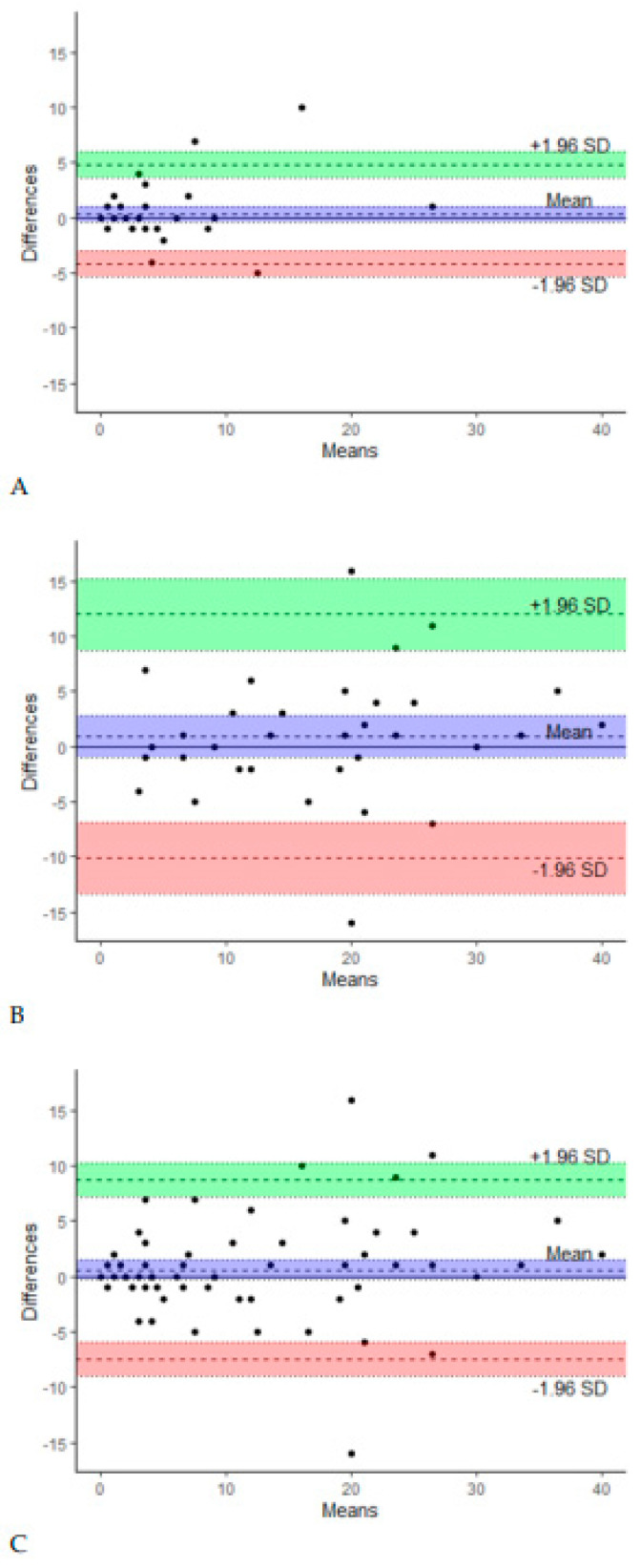
Bland–Altman plots of test and retest in group 0 (**A**), group 1 (**B**), and both groups combined (**C**).

**Table 1 cancers-16-02396-t001:** Demographic characteristics of study population.

Characteristic	Group 0, N = 50	Group 1, N = 50
Age BMI Lymphedema Grade 1	40 (SD 15), range 20–73 24.7 (SD 5.1), range 15.5–45.9 0 (NA%)	51 (SD 13), range 21–83 29.3 (SD 7.8), range 19.7–53.3 29 (58%)
Grade 2	0 (NA%)	21 (42%)

Group 0 is patient group without lower extremity lymphedema; group 1 is patient group with lower extremity lymphedema; BMI = body mass index; SD = standard deviation; NA = not applicable.

**Table 2 cancers-16-02396-t002:** Internal consistency of the Dutch validation version of the LELSQ.

Group	Cronbach’s Alpha	Cronbach’s Alpha Based in Standardized Items	N of Items
Group 0	0.835	0.852	11
Group 1	0.831	0.831	13
Group 0 + 1	0.909	0.91	13

**Table 3 cancers-16-02396-t003:** Summary of the face and content validity.

Questions in the Survey	Yes (n = 96)	No (n = 96)
Was the scoring system comprehensible	93 (96.9%)	3 (3%)
Was each question of the Dutch LELSQ understandable	95 (99%)	1 (1%)
Were all complaints related to your lymphedema questioned in the LELSQ	83 (86.46%)	13 (13.54%)

## Data Availability

The data associated with the paper are not publicly available but are available from the corresponding author on reasonable request.

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
