# Peer review of "Dutch Translation of the Yost Self-Report Lower Extremity Lymphedema Screening Questionnaire in Women"

_cancers, 2024, doi:10.3390/cancers16132396_

Round 1

Reviewer 1 Report (Previous Reviewer 1)

Comments and Suggestions for Authors

I have roughly reviewed the reply letter and revised manuscript, and the author responded to the reviewer's comments one by one, increased the sample size, and made serious and multiple revisions to the original manuscript. I agree to accept this manuscript.

Reviewer 2 Report (New Reviewer)

Comments and Suggestions for Authors

reference 19 and 11 are duplicated

This manuscript is a resubmission of an earlier submission. The following is a list of the peer review reports and author responses from that submission.

Round 1

Reviewer 1 Report

Comments and Suggestions for Authors

It is difficult to draw a stable and reliable conclusion due to insufficient sample size. I hope the authors can continue to expand the sample size to validate Dutch version of the screening questionnaire, the LELSQ. and then consider submitting the manuscript again.   

Dutch Translation of the Yost Self-Report Lower Extermity Lymphedema Screening Questionnaire in Women

Comments:

The authors translated and validated a Dutch version of the LELSQ developed by Yost in women. The validated Dutch translation showed high values for internal consistency, test-retest reliability as well as validity and face and content validity which allows the healthcare workers and researchers to implement the questionnaire in an early detection of LEL. This is an valuable study, because so far, no other patient questionnaire is available regarding the early detection of LEL in Dutch. The LELSQ can be used for early detecting of LEL, which permits to postpone or even prevent deterioration of the lymphedema. Early intervention may potentially have a positive impact on the quality of life of these patients.

There are some suggestion regarding this manuscript.

1. Line 12, please add City and Country.

2. Line 14,  Department Clinic ?

3. Line 21-25, The sentence is too long to understand, please revise it.

4. Line 33, Please specify which statistical analysis methods are used.

5. Introduction is insufficient, Please provide information on Lower-Extremity Lymphedema Screening Questionnaire in Belgium. Does Belgium not have such similar screening tools ?

6. Materials and Methods: Have you obtained authorization from Professor Kathleen Yost for translating and validating his/her questionnaire? Please supplement this content.

7. Although you used a formula to calculate the sample size, I think your sample size is too small. I hope you can further expand the sample size in future research to test the reliability and validity of this questionnaire.

8. Table 1, please add standard deviation SD.

9. Table 1 should also be supplemented with some other demographic characteristics, in addition to age and BMI. 

10. “ROC analysis showed an AUC value of 0.89 indicating a strong validity (P<0.001). The Youden index was 0.66 with a cut-off value of >6. This resulted in a sensitivity of 82.8% and a specificity of 83.3%“, due to  insufficient sample size, these yellow numbers are not very good,  I hope you can further expand the sample size to validate these indicators.

11. Table 3:  Were all complaints related to your lymphedema questioned in the LELSQ?    25 (83%)    5 (15%) ?      5(17%) , please revise it.

12. In the Discussion section, the comparisons between the Dutch  version of the questionnaire and other language versions need to be supplemented.  Although you mentioned the German version on line 311, the content is still insufficient.

13. Line 356, Insufficient sample size is also a limitation.

14. Please check the format of each reference, e.g., 21-22.   

Comments on the Quality of English Language

Moderate editing of English language required.   

Author Response

I hope the authors can continue to expand the sample size to validate the Dutch version of the screening questionnaire, the LELSQ and then consider submitting the manuscript again.

Reply:

An amendment to expand our sample size was submitted to the Ethics Committee in August 2023. Unfortunately, it was rejected due to the study’s established cutoff date. A completely new Ethics Committee was drafted and resubmitted for approval in October 2023. Approval was obtained from the Ethics Committee. Recruitment could now proceed to expand the study population. From December 2023 to April 2024 recruitment of the population was expanded to 100. This included both healthy individuals (without a diagnosis of lymphoedema) and patients with LEL (lymphoedema). The article has been revised with new statistical data based on the entire population of 100.

  • Line 12 : Thank you for pointing this out. I agree with this comment. Therefore, I have made the following change: City and Country are added: Ghent, Belgium.
  • Line 14 is corrected to Clinic for Lymphatic Disorders.
  • Line 21-25 Sentence is to long to understand, revise it: Here’s a revised version of the sentence to make it more concise and easier to understand: The validation of the questionnaire in Dutch was crucial for a larger trial called “The Gynolymph”, since there were no validated questionnaires in Dutch for the detection of early LEL. This trial aims to enroll four hundred patients who will utilize the screening tool as part of a patient-reported assessment. By doing so, the trial seeks to detect the early development of lymphedema after cancer treatment. To ensure unbiased results, the study team translated and adapted the lymphedema questionnaire to Dutch
  • Line 33 (now line 42-43) Statistical analyses SPSS (IBM corp, Armonk, New York) version 28.0.1.0 (001) was used for statistical analysis in the process of validation. The internal consistency was assessed by determining Cronbach’s alpha. Reliability was tested by a test-retest reliability. Validity was determined by ROC-analysis.
  • Introduction is insufficient, provide information on Lower-Extremity Lymphedema Screening Questionnaire in Belgium. Does Belgium not have similar screening tools?

Thank you for pointing this out. Therefore, I have made the following change since we did not have similar screening tools in Dutch:

However, as for now, there is no specific questionnaire available for screening lymphedema in the lower limbs in Dutch. We currently possess the validated Dutch-language Lymphedema Functioning, Disability, and Health Questionnaire (lymph-ICF). This questionnaire serves as a reliable measurement tool to assess the quality of life for patients with lymphedema in the lower limbs, but not for early detection of LEL. The primary objective of our study was to validate and translate a Dutch version of the validated Lower-Extremity Lymphedema Screening Questionnaire (LELSQ) developed by Yost.

  • Materials and Methods: The authorization from Kathleen Yost was sent to the editor. I have adapted in text as follow : “For this study, the LELSQ was used, as developed, and validated by Yost et al., upon written approval for copyright usage.”

Dr. Monten has my permission to prepare a separate Dutch translation of the questionnaire. I can provide more detailed guidance on translation methodology, if needed. As a reminder, Mayo Clinic retains the copyright on any derivatives of the original questionnaire, including translations. Thank you both again for your interest in the Lower Extremity Lymphedema screening questionnaire. Please let me know if you have any questions or concerns about this message. Sincerely,Kathleen

Kathleen Yost, PhD | Professor of Health Services Research |Director, Survey Research Center | Affiliate, Kern Center for the Science of Health Care Delivery

Mayo Clinic | 200 First Street SW | Rochester MN 55905

(507) 538-3894

  • Sample size too small: Sample size is adapted to 100. ROC analysis adapted after sample size adaptation. In the text study population is changed to : “Due to the diversity of questionnaire types, there are no universally applicable rules regarding the required sample size for questionnaire validation. While larger sample sizes are generally preferable, there is no one-size-fits-all rule for determining the necessary sample size for questionnaire validation. Additionally, considering the respondent-to-item ratios can provide further justification for a larger sample size when needed. Number of volunteers / patients needed per group was calculated through sample size analysis, imposing a power of at least 90%  and based on the formula by Bonett. There are 13 items in the questionnaire of which the reliability of its measurements needs to be measured.  According to this formula the sample size calculation we needed was a minimum of 13 persons included. Another guideline for the respondent-to-item ratio is the 5:1 rule. The rule suggests that for every item in the questionnaire, we aim for at least 5 respondents. A minimum sample size of a total of 65 respondents is required for the 13-item questionnaire based on the recommended respondent-to-item ratio of 5:1. As a test-retest was foreseen within 2 weeks after the first completion, the number of participants was increased to 100 to compensate potential drop out.”
  • Table 1: standard deviation is added.
  • Table 3: is revised.
  • German version : We requested permission for the German version as you suggested by email in January 2024, but to date, we have not received any response.
  • Line 356 sample size is adapted up to 100.
  • References format adapted.

Reviewer 2 Report

Comments and Suggestions for Authors

A Lower Extremity Lymphoedema Screening Questionnaire in women was developed, a simple and easy-to-use screening tool for early detection of Lower Extremity Lymphoedema (LEL).

A cross-sectional investigation is recommended, followed in the report, since it aims to collect participant data at a specific time. Reliability analysis, Validity assessment, Factor analysis, Pilot testing, etc., could be used to assess the quality and reliability of the questionnaire.

The paper follows all these rules.

·        Sample size selection

Sample size determination for the number of participants needed for the study is essential. A sufficient sample size is needed so that the research can provide reliable and reproducible evidence that can detect an instrument’s desired consistency or stability. The power is set at 90%, and expected Cronbach’s alpha at 0.75. The paper follows this, but again ‘Check the calculation of the formula.’ n = 11.66, approximately equal to 12

·        Internal consistency

Cronbach’s alpha is used to measure internal consistency. The range is 0-1. The values were between 0.81 and .9, which shows a strong correlation.

The two questions in the questionnaire show no added value in the non-lymphoedema group, and their removal from the entire questionnaire may be considered without significantly impacting the reliability. Alternatively, you could suggest that those two questions can be optional in the Dutch translation because questionnaire studies are often culture-specific.

·        Test-retest reliability

ICC calculated. Strong to moderate results are obtained. Figure 2 is indicated as Figure 1 in the heading of the plot.

·        VALIDITY

It shows a strong criterion validity, indicated by a high AUC value and optimal cut-off value, and sensitivity and specificity greater than 80%.

One relevant reference for consideration of the authors

Comparison of Three Quality of Life Instruments in Lymphatic Filariasis: DLQI, WHODAS 2.0, and LFSQQ PLoS Negl Trop Dis 2014; 8: e2716. doi:10.1371/journal.pntd.0002716

Author Response

Comments Reviewer 2

Thank you very much for reading and reviewing the article. I have provided the necessary adjustments to the manuscript based on your comments.

Internal Consistency

The two questions are only answered 0 in the non-lymphedema group. Patients with lymphedema can suffer from genital edema and therefore we decided to let the questions in the questionnaire, since the question is not culture specific.

Line 251 is adjusted to Figure 1 in the text instead of Figure 2, is now line 305.

Validity

In our study, we referred to the suggested article on validity. Specifically, in line 442, we explained that we couldn’t conduct discriminant validity due to the absence of other Dutch questionnaires validated for detecting lower extremity oedema. Therefore, it was not possible to make a comparison like in the suggested study.